# Influence of body position on microcirculatory and autonomic responses during arterial occlusion in healthy adults

Robert Trybulski[1,2*], Adrian Kużdzał[3], Gabriel Stanica Lupu[4], Wacław Kuczmik[5], Grzegorz Biolik[6], Magdalena Hagner Derengowska[7], Jakub Taradaj[8]

1 Medical Department Wojciech Korfanty, Upper Silesian Academy in Katowice, Katowice, Poland, 2 Provita Żory Medical Center, Żory, Poland, 3 Institute of Physiotherapy, Faculty of Health Sciences and Psychology, Collegium Medicum, University of Rzeszów, Poland, 4 Faculty of Movement, Sports and Health Sciences, "Vasile Alecsandri" University of Bacău, Bacău, Romania, 5 Department and Clinic of General Surgery, Vascular Surgery, Angiology and Phlebology, Faculty of Medical Sciences in Katowice, Medical University of Silesia in Katowice, Katowice, Poland, 6 Department of General Surgery, Vascular Surgery, Angiology and Phlebology, Medical University of Silesia, Katowice, Poland, 7 Faculty of Earth Sciences and Spatial Management Nicolaus Copernicus University, Torun, Poland, 8 Institute of Physiotherapy and Health Sciences, Academy of Physical Education in Katowice, Katowice, Poland

* roberttrybulski@proton.me

## Abstract

This study aimed to investigate how body position (supine, seated, and standing) influences post-occlusive reactive hyperemia (PORH) and autonomic nervous system activity, with implications for vascular health and the safety of blood flow-restricted (BFR) exercise. A prospective, within-subject design was implemented. Fifteen healthy participants (18–30 years) were evaluated across three experimental sessions. Each session involved one randomized body position. Arterial occlusion pressure (AOP), microcirculatory responses (resting flow [RF], biological zero [BZ], time to peak [TP], recovery time [TR]), and heart rate variability (HRV) parameters—average normal-to-normal interval (AVNN), standard deviation of normal-to-normal intervals (SDNN), low-frequency/high-frequency power ratio (LF/HF), and heart rate (HR)—were recorded using laser Doppler flowmetry, Doppler ultrasound, and Polar H10 chest sensors. Standing and sitting positions significantly increased AOP min and AOP 100% compared to supine (p = 0.003, r ≥ 0.62). Resting flow and BZ values were also significantly elevated in these upright positions (sitting and standing) (p = 0.003). Time to peak and recovery time were longest in the standing position (p = 0.003–0.009). HRV analyses revealed a significant decrease in AVNN from supine to sitting and standing (p = 0.003), while SDNN and LF/HF increased in upright positions (sitting and standing) (p < 0.013). Heart rate was significantly higher in standing vs. sitting (p = 0.006), but not between supine and sitting (p = 0.459). Body position markedly influences both microcirculatory dynamics and autonomic nervous system responses. Upright positions (sitting and standing) elevate AOP and alter

**Data availability statement:** The dataset used in this research includes information that could be sensitive or indirectly identify participants. For ethical and legal reasons, it cannot be made publicly available, even in de-identified form, to ensure the protection of participant privacy. While all individuals involved gave written consent to take part in the study, their approval did not extend to open sharing of their communication data. Scholars who wish to replicate the findings or conduct secondary analyses may request controlled access by submitting a formal application to the Medical Department of the Wojciech Korfanty Upper Silesian Academy in Katowice, 40-659 Katowice, via wydzial. medyczny@gwsh.pl.

**Funding:** The author(s) received no specific funding for this work.

**Competing interests:** The authors have declared that no competing interests exist.

HRV, suggesting increased vascular and sympathetic activity. These findings are critical for optimizing BFR protocols and assessing cardiovascular safety during postural changes.

## 1. Introduction

The current literature reveals important gaps in understanding how body position influences post-occlusion reactive hyperemia (PORH)—a physiological response essential for evaluating microvascular function. This has particular relevance in cardiovascular research and sports medicine, where blood flow restriction training (BFRT) is gaining prominence [1]. The microcirculatory system, composed of vessels under 150 μm, including arterioles, capillaries, venules, and arteriovenous shunts, plays a central role in tissue perfusion and the exchange of gases and nutrients. Its ability to adapt to ischemic events is crucial in minimizing tissue damage [2].

PORH describes the surge in blood flow following the release of a temporary occlusion, serving as a marker for endothelial health. Its regulation involves multiple mechanisms, including the axonal reflex and endothelium-derived factors such as nitric oxide (NO) and endothelium-derived hyperpolarizing factor (EDHF) [3,4]. NO in particular facilitates vasodilation through the cGMP pathway, especially in skeletal and muscular vasculature [5]. Body posture can significantly impact PORH by altering venous return, arterial inflow, and vascular resistance, which in turn influence perfusion pressure and vascular responses [6,7].

These physiological shifts are especially relevant in BFRT protocols, which are commonly performed in supine, seated, or standing positions. In the present study, we refer to seated and standing collectively as 'upright positions,' to distinguish them from the supine posture. While BFRT methods such as ischemic preconditioning (IPC) and partial occlusion during exercise are well documented [8], the effects of posture on microcirculatory and autonomic responses remain underexplored [9]. BFRT-induced muscle adaptations are largely driven by hypoxia and metabolite accumulation, enhancing muscle activation and anabolic signaling. These effects are modulated by arterial occlusion pressure (AOP), which varies by limb size and body composition [10,11].

Posture not only affects AOP but also influences endothelial shear stress, impacting nitric oxide (NO) release and vascular conductance [12,13]. Furthermore, the autonomic nervous system (ANS), which governs cardiovascular regulation, responds dynamically to both exercise and changes in posture. Variations in heart rate variability (HRV) metrics reflect shifts in sympathetic and parasympathetic activity across different positions and exercise intensities [9,14,15].

From a clinical perspective, posture-dependent changes in PORH and vascular conduction are significant, particularly in rehabilitation and preoperative settings. For example, AOP tends to be lower in the supine position, which may result in underestimation of the occlusion threshold if these values are later applied during upright exercise [9,16]. Such miscalibration can compromise BFRT efficacy (by providing

insufficient stimulus) or increase cardiovascular risk (if pressures are inappropriately elevated). Clarifying these posture-related effects is therefore critical for optimizing BFRT protocols and ensuring both safety and effectiveness [9,16]. Research also suggests that limb position affects PORH magnitude, with horizontally or downward-oriented limbs showing stronger hyperemic responses, likely due to hydrostatic pressure effects [4,17]. Beyond limb positioning, posture and unloading strategies also influence cutaneous microcirculation. In individuals with spinal cord injury, durations of wheelchair tilt-in-space and recline significantly alter skin perfusion over pressure-bearing regions [18]. Likewise, reviews of skin blood flow dynamics emphasize how positional changes contribute to ischemia–reperfusion responses and tissue vulnerability [19].

HRV provides a non-invasive index of ANS balance, reflecting the dynamic interplay between sympathetic and parasympathetic inputs to the heart [20]. Given that the ANS also regulates vascular tone, particularly through sympathetic vasoconstrictor activity, HRV measures can provide indirect insight into microvascular responses such as PORH [21]. Previous work indicates that autonomic shifts influence endothelial function and cutaneous vasodilation, with sympathetic activation attenuating hyperemic responses and parasympathetic predominance facilitating more efficient recovery [22]. Therefore, evaluating HRV in parallel with PORH allows for a more comprehensive assessment of how posture influences both vascular and autonomic components of circulatory regulation.

Acknowledging the existing knowledge gaps and considering the aforementioned points, this study endeavors to clarify the complex relationships among body position, PORH, and autonomic nervous system activity in a cohort of healthy individuals. This investigation holds significant implications for enhancing our understanding of vascular health and the safety considerations associated with blood flow-restricted training protocols. By analyzing microcirculatory responses—arterial occlusion pressure (AOP), resting flow (RF), time to peak (TP), recovery time (TR), and biological zero (BZ)—together with autonomic activity assessed via heart rate variability (HRV) parameters [average NN interval (AVNN), standard deviation of NN intervals (SDNN), and low-frequency/high-frequency ratio (LF/HF), this study aims to determine how lower limb positions (supine, sitting, and standing) influence microcirculatory hemodynamic and autonomic responses.

## 2. Methods

### 2.1. Experimental design

This investigation employed a prospective experimental design lasting between 13/03/2024 to 10/06/2024. Within this framework, participants underwent testing in three distinct body positions – supine, seated, and standing – while under conditions of arterial occlusion.

### 2.2. Trial registration

Following a thorough explanation of the potential risks and benefits associated with the experimental procedures, all participants provided their written informed consent. They were also informed of their right to withdraw from the study at any point without consequence. This research received approval from the ethical committee of the Polish Society of Physiotherapy (reference number: 3.03.2024) and was registered and published as a clinical trial under the number ISRCTN15418049 (18/04/2024) [23]. The study was conducted in accordance with the ethical principles outlined in the Declaration of Helsinki.

### 2.3. Experimental setting

Participants visited the Provita Medical Center on three occasions, scheduled between Monday and Wednesday, specifically between 9:00 and 11:00 a.m. Upon arrival, initial anthropometric data were collected using a Tanita MC-580 P bioimpedance body composition analyzer (Japan, 2022). Subsequently, participants were instructed to rest in a seated position for a duration of 20 minutes. On the following days, volunteers adopted the supine, sitting, and standing positions for the experimental procedures. Prior to the commencement of data collection, they remained in each of these positions

for 5 minutes. The measurement positions were implemented following established scientific literature guidelines to ensure the stability of the Laser Doppler Flowmetry (LDF) measurements [24,25]. All assessments were conducted by the same pair of researchers, a physician and a physiotherapist, within a temperature-controlled environment (temperature: 21.2±0.75 °C; humidity: 54.49±2.5%).

## 2.4. Participants and eligibility criteria

Inclusion criteria were as follows: (1) healthy volunteers aged 18–30 years; (2) classified as level 1 according to McKay's participant classification scheme [26]; (3) non-smokers; (4) ankle-brachial index (ABI) within the range of 0.9 and 1.4; (5) absence of any diagnosed health conditions; (6) not taking any medications; (7) abstinence from vigorous physical activity and alcohol for 24 hours prior to the study; (8) no consumption of ergogenic drinks (e.g., coffee, cola) for six hours before participation; and (9) female participants assessed during the follicular phase of their menstrual cycle. Exclusion criteria included: (1) blood pressure equal to or exceeding 140/95 mmHg; (2) use of steroids or hormonal contraceptives; (3) nicotine addiction; and (4) use of medications affecting systemic hemodynamics, such as β-blockers, calcium antagonists, or renin-angiotensin system inhibitors.

Participants were recruited through convenience sampling from a population of healthy young adults within the age range of 18–30 years (Fig 1). Recruitment was conducted via public announcements and direct invitations at local universities and community centers. Interested individuals were initially screened through a brief health questionnaire to assess eligibility based on inclusion and exclusion criteria. Those meeting the criteria were invited to participate in the study and provided with detailed information regarding the procedures, risks, and benefits.

## 2.5. Experimental conditions

Participants were assessed in three distinct body positions—supine, seated, and standing—chosen to evaluate the potential influence of posture on physiological responses during arterial occlusion. In the supine position, participants lay flat on their backs with arms comfortably at their sides. In the seated position, they sat upright on a chair with feet flat on the floor and back supported. In the standing position, participants stood upright with feet shoulder-width apart, maintaining a relaxed posture without shifting weight. Each position was tested on a separate day over three consecutive days. To control for potential order effects and ensure balanced exposure, the sequence of positions was randomly assigned using sealed opaque envelopes. Specifically, 5 participants completed supine first, followed by seated and then standing; 4 participants began with seated, then standing, and finally supine; and 6 participants started with standing, followed by supine, and then seated.

## 2.6. Outcomes and measurements

Following a 10-minute rest in the supine position, participants underwent a series of standardized measurement procedures conducted over three consecutive days, with each session focusing on a different body position—supine, seated, or standing—in a varied sequence across participants. Initial assessments included thigh circumference measurement and femoral artery depth evaluation using ultrasound to personalize occlusion pressure. Polar H10 chest strap sensors were then applied to record heart rate variability (HRV), followed by the determination of the ankle-brachial index (ABI) using Doppler ultrasound. Subsequently, a laser Doppler flowmeter (LDF) probe was positioned on the big toe to monitor microcirculation responses, and arterial occlusion was induced with a pneumatic cuff to identify individual arterial occlusion pressure (AOP) (Fig 2).

Each pressure step was held briefly to stabilize blood flow, and post-occlusion reactive hyperemia (PORH) was assessed. These procedures allowed for comprehensive, position-specific evaluations of vascular and autonomic responses relevant to blood flow restriction protocols.

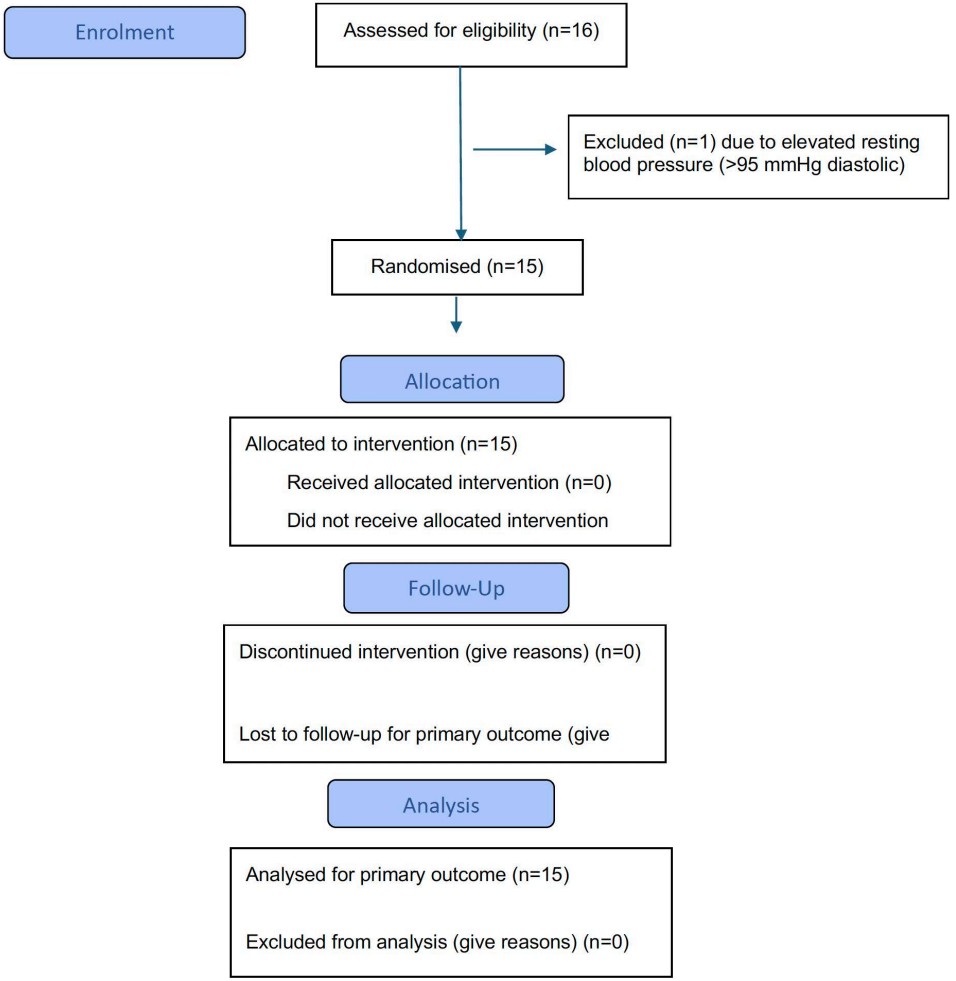

**Fig 1. CONSORT 2025 Flow Diagram [70].**

Below is the sequence of assessments and conditions:

1. In a resting supine position, thigh circumference was measured using a tape measure, and the femoral artery depth at the cuff location was assessed with ultrasound. Thigh circumference plays a crucial role in determining the pressure needed to occlude the artery, especially during blood flow restriction (BFR) exercise. Variables like cuff width and body posture impact the relationship between thigh circumference and arterial occlusion pressure (AOP). Gaining insight into these factors is essential for safely and effectively prescribing BFR exercises in both fitness and therapeutic settings [27].

2. HRV was recorded continuously using a Polar H10 chest strap (Polar Electro Oy, Finland) and stored via the HRV Logger application. R-R intervals were sampled at 1000 Hz and exported in raw format. Each participant underwent a standardized 5-minute recording in the designated body position (supine, seated, standing) following a 10-minute stabilization period. HRV serves as a non-invasive indicator of autonomic nervous system function, reflecting the balance between the sympathetic and parasympathetic nervous systems' influences on heart rate [28]. Body position can notably shift this balance, affecting HRV measurements. Studies suggest that specific HRV indices reflecting

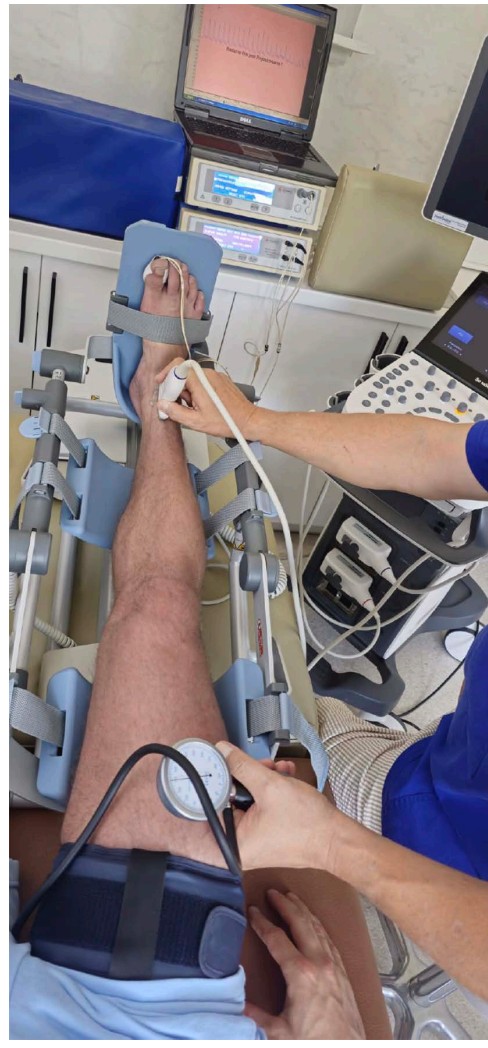

**Fig 2. Leg Stabilization During Measurements arterial occlusion pressure.**

parasympathetic activity — such as higher AVNN, greater SDNN, and increased high-frequency (HF) power — are more pronounced in the supine position [29,30]. In contrast, sitting or standing postures elicits a shift toward sympathetic dominance, reflected in shorter AVNN, reduced HF power, increased LF/HF ratio, and higher heart rate [31]. This has relevance in blood flow restriction training (BFRT) and its effects on hyperemic rebound [32]. Frequency-domain indices were calculated using Fast Fourier Transform (FFT) spectral analysis with Welch's method (window length: 256 s, 50% overlap, Hanning window). All HRV indices were computed using Kubios HRV Premium (Kubios Oy, Finland), which is widely used for clinical and sports applications. The primary outcomes analyzed across positions were median [IQR] values for AVNN (mean of all normal-to-normal R-R intervals, representing overall heart period), SDNN (standard deviation of all NN intervals, reflecting total HRV), LF/HF (ratio of low-frequency (0.04–0.15 Hz) to high-frequency (0.15–0.40 Hz) power, indicating sympathovagal balance), and HR.

3. An experienced physician utilized a SonoScape P20 B-mode ultrasound system (SonoScape, China, 2022) equipped with a linear-array transducer (4–18 MHz) to measure the ankle–brachial index (ABI). The dominant leg, determined

by self-reported kicking preference, was assessed. Systolic pressures were obtained at the brachial artery and at the ankle using both the posterior tibial and dorsalis pedis arteries. The higher of the two ankle pressures was used for ABI calculation. To ensure consistency, the probe position was marked on the skin after application of ultrasound gel [33].

4. The next step involved disinfecting the skin of the big toe, followed by placement of the Laser Doppler Flowmeter (LDF) probe on the plantar side of the toe, aligning it with the center of the nail.

5. The physician monitored the tibial pulse of the dominant leg, continuing until arterial flow was fully occluded. The Doppler sensor was placed over both arteries, ensuring a consistent insonation angle of less than 60º [34]. The Doppler gate was positioned to capture the entire vessel. Standardization of ABI measurements is essential for accurate diagnosis and assessment of peripheral arterial disease (PAD) and cardiovascular risk, as well as for the safe application of BFRT [35]. Arterial Occlusion Pressure (AOP) was determined individually for each participant on the dominant leg. A 13-cm wide pneumatic cuff (Riester®, Germany) was positioned proximally at the inguinal crease. A SonoScape P20 B-mode ultrasound system (SonoScape, China, 2022) equipped with a linear-array transducer (4–18 MHz) was used in pulsed Doppler mode, with the probe placed over the posterior tibial and dorsalis pedis arteries to continuously monitor arterial flow. Cuff inflation began at 0 mmHg and increased in 10 mmHg increments after an initial rise to 100 mmHg. At each step, pressure was held for 30 seconds to allow flow stabilization, while Doppler confirmed vessel patency [11]. The AOP was defined as the cuff pressure at which arterial flow was no longer detectable by Doppler. For analysis, we recorded both the minimal pressure reducing resting flow (AOPmin) and the pressure causing complete occlusion (AOP100). This individualized approach accounts for inter-subject differences in limb circumference, vessel depth, and posture, consistent with published BFR protocols. The study employed a standard 5-minute occlusion test on the dominant leg [36].

6. The PeriFlux System 5000 Laser Doppler Flowmeter (Perimed AB, Järfälla, Sweden, 2004) was used to evaluate post-occlusion responses. As the gold standard for assessing microcirculatory responses, LDF provides highly sensitive and reproducible measurements [24]. The reflected wave from erythrocytes was recorded from a 1 mm³ volume of skin tissue at a depth of 2.5 mm, with a sampling frequency of 32 Hz. A contact laser probe was used to capture the data. This technique employs monochromatic laser light in the red to near-infrared spectrum [37]. The device projects a beam deep into tissue, where photons interact with moving blood cells, altering their frequency due to the Doppler effect. The resulting flux corresponds to the velocity and concentration of erythrocytes in the area, which is influenced by skin vasoconstriction and vasodilation [38]. A photodetector system analyzes the reflected light, converting it into a voltage proportional to the erythrocyte velocity and quantity [25]. Raw flux signals were processed offline with a 5-s moving average filter to minimize high-frequency noise. Resting flow (RF) was defined as the mean flux during the 60 s immediately prior to cuff inflation. Peak perfusion (RHmax) was identified as the highest filtered flux value within 2 min of cuff release, and time to peak (TP) was measured as the interval between cuff release and RHmax. Recovery time (TR) was defined as the time required for flux to return to within 5% of baseline RF.

The PORH study utilized well-established, standardized measures to assess microcirculation responses (Fig 3) [25,39], including:

- RF Rest Flow: Perfusion unit (no reference provided) [PU]

- TP (Time to Peak): The time from the end of occlusion to peak perfusion [s]

- TR (Recovery Time): Time taken for perfusion to return to baseline after peak hyperemia [min]

- AOP min: The lowest pressure where resting flow is reduced [mmHg]

- AOP 100%: The pressure for complete arterial occlusion [mmHg]

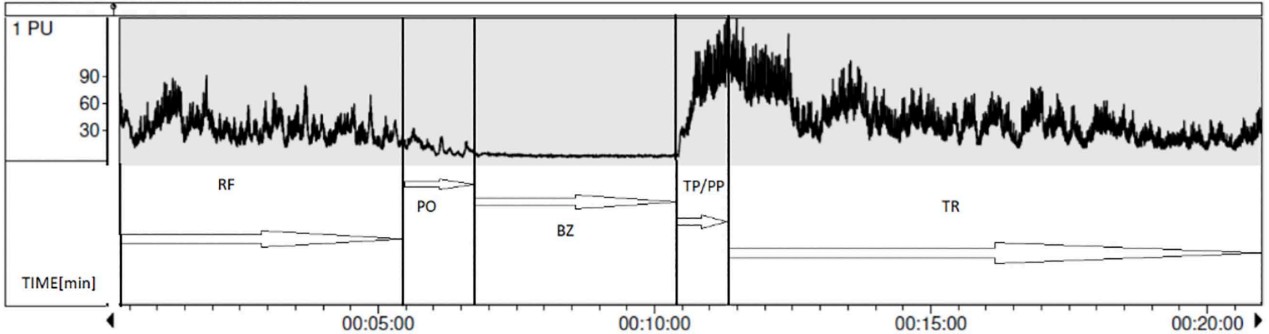

**Fig 3. Example of LDF recording during post-occlusive reactive hyperemia (PORH).** **RF-** rest flow; **PO-** pressure occlusive -the lowest recorded pressure at which there is no flow in the dorsal tibial artery; **TP –** Time to peak – time from the end of induced occlusion to the maximum value of perfusion; **PP** – peak perfusion (Max) – maximum value of perfusion; **TR** – recovery time – time needed for perfusion to return to the resting value after reaching the peak value of hyperemia.

- BZ (Biological Zero): Flow value during total occlusion [PU]

HRV analysis, particularly its components, is a non-invasive method for evaluating autonomic nervous system activity, offering insights into autonomic balance after interventions like BFR [28,40]. Key HRV components include:

- AVNN (Average NN Interval): The average time between successive heartbeats, measured in milliseconds. Longer intervals indicate a lower heart rate, while shorter intervals signal a higher rate. Normal values range from 800–1000 ms. Body position affects AVNN, as sympathetic activity tends to be higher in upright positions, influencing heart rate [32].

- SDNN (Standard Deviation of NN Intervals): This measures total heart rate variability, considering both sympathetic and parasympathetic influences. For healthy adults, SDNN norms are: low variability <20 ms, medium 20–50 ms, and high variability >50 ms [41].

- LF/HF Ratio: The ratio of low-frequency (LF) power to high-frequency (HF) power, used to assess sympathetic-parasympathetic balance. Norms: <1 indicates parasympathetic dominance (relaxation), 1–3 suggests balance, and >3 indicates sympathetic dominance (stress) [42].

## 2.7. Sample size

Sample size was estimated a priori using G*Power 3.1. Since no directly comparable studies combining posture, PORH, and HRV outcomes were available at the time of planning, we adopted a medium effect size (f = 0.25) as recommended by Cohen [43] for ANOVA-type designs. This approach balances feasibility with statistical power when prior data are lacking. With $\alpha = 0.05$ and power (1-$\beta$) = 0.80, the required sample size was 15 participants.

## 2.8. Randomization

Randomization of the three body positions (supine, seated, and standing) was performed within each participant using a computer-generated random sequence. This ensured that the order in which each body position was assessed was randomized for every participant, eliminating potential bias from the sequence of conditions. By having the same participant undergo all three conditions, we were able to control for individual differences, ensuring that the effects observed were due to the body position rather than variations between participants.

## 2.9. Statistical methods

Given the small sample size (n = 15) and non-normality of the data, nonparametric tests were used to analyze the data. Normality of each variable was assessed using the Shapiro–Wilk test (p < 0.05). The Friedman test was conducted to assess differences between the three body position conditions (supine, seated, and standing). When the Friedman test indicated significant differences, pairwise comparisons were performed using the Wilcoxon signed-rank test to identify which specific conditions differed. To control for the increased risk of Type I error, the Bonferroni correction was applied to the Wilcoxon results, and these Bonferroni-adjusted p-values are presented in the main tables. As an additional check, we also calculated exact Wilcoxon p-values with Holm correction across the three pairwise comparisons per outcome.

To quantify effect sizes, two complementary measures were reported. First, Kendall's W was calculated for the Friedman test as a robust global effect size that is particularly appropriate for small-sample nonparametric designs. W values range from 0 (no agreement) to 1 (complete agreement), with 0.1, 0.3, and 0.5 often interpreted as small, moderate, and large effects, respectively. Second, the effect size r was calculated for Wilcoxon signed-rank tests as Z divided by the square root of the sample size. Given the limitations of r with small samples (normal approximation), these values are provided as descriptive indicators of magnitude.

In addition, exploratory analyses were conducted to assess potential sequence (order) effects, that is, whether outcomes differed depending on whether a given condition was performed in the first, second, or third session. For this purpose, Kruskal–Wallis tests were applied separately to each primary outcome variable. Statistical procedures were conducted using SPSS software (version 28.0.0., IBM, USA), with significance set at p < 0.05.

## 3. Results

Among the recruited participants, 15 healthy, physically active individuals were eligible for this study, including 9 men and 6 women. The average age of the participants was 26.6 ± 3.1 years, with an average height of 173.5 ± 11.8 cm, body mass of 68.4 ± 15.3 kg, and a body mass index (BMI) of 23.4 ± 1.6 kg/m². The average ankle-brachial index (ABI) was 1.02 ± 0.04, while systolic blood pressure averaged 127.7 ± 2.6 mmHg and diastolic blood pressure was 80.3 ± 2.5 mmHg.

The results of the Friedman test indicated that there were significant differences between conditions in rest flow ($\chi^2(2)$ = 30.000, p < 0.001, W = 1.00), AOPmin ($\chi^2(2)$ = 28.526, p < 0.001, W = 0.95), AOP100% ($\chi^2(2)$ = 29.000, p < 0.001, W = 0.97), biological zero (BZ) ($\chi^2(2)$ = 30.000, p < 0.001, W = 1.00), reactive hyperemia (RHmax) ($\chi^2(2)$ = 25.200, p < 0.001, W = 0.84), time to peak ($\chi^2(2)$ = 25.864, p < 0.001, W = 0.86), recovery time (TR) ($\chi^2(2)$ = 29.525, p < 0.001, W = 0.98), AVNN ($\chi^2(2)$ = 23.333, p < 0.001, W = 0.78), SDNN ($\chi^2(2)$ = 26.533, p < 0.001, W = 0.89), LF/HF ratio ($\chi^2(2)$ = 29.525, p < 0.001, W = 0.98), and HR ($\chi^2(2)$ = 16.877, p < 0.001, W = 0.56). Table 1 presents the descriptive statistics for the outcomes across the three conditions and the comparisons between conditions using the Wilcoxon and the adjusted Bonferroni.

Fig 4 exhibits mean values and 95% confidence intervals for resting flow, AOP, biological zero, reactive hyperemia (RHmax), and time to peak across the three conditions. Standing position yielded significantly greater resting flow compared to the supine (mean difference: 2.0 PU; p = 0.003; r=−0.62) and sitting positions (mean difference: 0.9 PU; p = 0.003; r=−0.62). Furthermore, sitting also resulted in significantly higher resting flow than supine (mean difference: 1.1 PU; p = 0.003; r=−0.64).

Standing position showed significantly higher AOP min compared to supine (mean difference: 11.4 mmHg; p = 0.003; r=−0.64) and sitting (mean difference: 6.4 mmHg; p = 0.003; r=−0.62). Sitting also had significantly higher AOP min than supine (mean difference: 5.0 mmHg; p = 0.003; r=−0.62). Standing yielded significantly greater AOP 100% than both supine (mean difference: 14.6 mmHg; p = 0.003; r=−0.63) and sitting positions (mean difference: 7.6 mmHg; p = 0.003; r=−0.62). Sitting also resulted in significantly higher AOP 100% than supine (mean difference: 7.0 mmHg; p = 0.003; r=−0.62).

Standing position exhibited significantly higher BZ values than both supine (mean difference: 1.88 PU; p = 0.003; r=−0.62) and sitting (mean difference: 1.10 PU; p = 0.003; r=−0.62). Additionally, sitting showed significantly higher BZ

**Table 1. Means, standard deviations, and inferential statistics for the outcomes in the three conditions tested.**

| Condition | Supine (H) | Sitting (Sit) | Standing (Sta) | Sit vs. H | Sta vs. H | Sta vs. Sit |
|---|---|---|---|---|---|---|
| Rest flow (PU) mean±SD | 12.1±1.6 | 13.2±1.5 | 14.1±1.5 | Z=−3.495; p=0.003*; r=−0.64; Holm p=0.0001 | Z=−3.423; p=0.003*; r=−0.62; Holm p=0.0001 | Z=−3.436; p=0.003*; r=−0.62; Holm p=0.0001 |
| Median [IQR] | 12.20 [10.95–13.00] | 13.20 [12.20–14.15] | 14.20 [13.45–14.85] | | | |
| AOP min (mmHg) mean±SD | 125.3±7.4 | 130.3±8.3 | 136.7±6.7 | Z=−3.419; p=0.003*; r=−0.62; Holm p=0.0006 | Z=−3.508; p=0.003*; r=−0.64; Holm p=0.0002 | Z=−3.450; p=0.003*; r=−0.62; Holm p=0.0006 |
| Median [IQR] | 125.00 [120.00–130.00] | 130.00 [125.00–135.00] | 135.00 [130.00–142.50] | | | |
| AOP 100% (mmHg) mean±SD | 174.7±5.8 | 181.7±4.1 | 189.3±5.0 | Z=−3.407; p=0.003*; r=−0.62;; Holm p=0.0007 | Z=−3.438; p=0.003*; r=−0.63;; Holm p=0.0002 | Z=−3.416; p=0.003*; r=−0.62;; Holm p=0.0007 |
| Median [IQR] | 175.00 [170.00–180.00] | 185.00 [180.00–185.00] | 190.00 [185.00–195.00] | | | |
| BZ (PU) mean±SD | 3.45±0.39 | 4.23±0.46 | 5.33±0.52 | Z=−3.451; p=0.003*; r=−0.63; Holm p=0.0001 | Z=−3.415; p=0.003*; r=−0.62; Holm p=0.0001 | Z=−3.423; p=0.003*; r=−0.62; Holm p=0.0001 |
| Median [IQR] | 3.50 [3.15–3.75] | 3.90 [3.90–4.75] | 5.50 [4.80–5.80] | | | |
| RHmax (PU) mean±SD | 69.9±15.7 | 75.4±14.7 | 82.6±13.7 | Z=−3.069; p=0.006*; r=−0.56; Holm p=0.0009 | Z=−3.408; p=0.003*; r=−0.63; Holm p=0.0001 | Z=−3.417; p=0.003*; r=−0.63; Holm p=0.0001 |
| Median [IQR] | 76.70 [59.15–81.75] | 79.40 [67.50–84.50] | 88.10 [81.25–89.45] | | | |
| TR (min) mean±SD | 7.70±0.90 | 9.70±0.74 | 17.81±23.87 | Z=−3.433; p=0.003*; r=−0.63; Holm p=0.0001 | Z=−3.409; p=0.003*; r=−0.63; Holm p=0.0001 | Z=−3.324; p=0.003*; r=−0.61; Holm p=0.0009 |
| Median [IQR] | 7.70 [6.85–8.50] | 9.40 [9.20–10.25] | 12.10 [10.40–12.85] | | | |
| Time to peak (s) mean±SD | 20.1±3.5 | 24.9±3.5 | 31.1±3.8 | Z=−2.993; p=0.009*; r=−0.56; Holm p=0.0015 | Z=−3.417; p=0.003*; r=−0.63; Holm p=0.0002 | Z=−3.300; p=0.003*; r=−0.61; Holm p=0.0015 |
| Median [IQR] | 21.00 [17.00–22.50] | 26.00 [22.50–27.50] | 32.00 [28.50–34.00] | | | |
| AVNN (ms) mean±SD | 907.7±34.6 | 750.7±33.6 | 729.6±48.7 | Z=−3.408; p=0.003*; r=−0.63; Holm p=0.0001 | Z=−3.408; p=0.003*; r=−0.63; Holm p=0.0001 | Z=−1.933; p=0.159; r=−0.35; Holm p=0.0554 |
| Median [IQR] | 921.00 [883.00–933.50] | 754.00 [733.50–762.50] | 733.00 [714.50–751.00] | | | |
| SDNN (ms) mean±SD | 34.3±4.4 | 53.3±4.4 | 59.1±2.7 | Z=−3.410; p=0.003*; r=−0.63; Holm p=0.0001 | Z=−3.411; p=0.003*; r=−0.63; Holm p=0.0001 | Z=−2.882; p=0.012*; r=−0.53; Holm p=0.0554 |
| Median [IQR] | 35.00 [31.50–37.50] | 52.00 [49.00–57.00] | 59.00 [58.00–59.00] | | | |
| LF/HF (A.U.) mean±SD | 1.37±0.13 | 1.83±0.12 | 2.05±0.11 | Z=−3.438; p=0.003*; r=−0.63; Holm p=0.0001 | Z=−3.423; p=0.003*; r=−0.62; Holm p=0.0001 | Z=−3.359; p=0.003*; r=−0.62; Holm p=0.0009 |
| Median [IQR] | 1.40 [1.30–1.40] | 1.80 [1.75–1.90] | 2.10 [1.95–2.10] | | | |
| HR (bpm) mean±SD | 71.1±9.8 | 72.1±8.6 | 82.1±3.7 | Z=−1.429; p=0.459; r=−0.26; Holm p=0.1531 | Z=−3.042; p=0.006*; r=−0.56; Holm p=0.0023 | Z=−3.125; p=0.006*; r=−0.57; Holm p=0.0018 |

*(Continued)*

**Table 1.** (Continued)

| Condition | Supine (H) | Sitting (Sit) | Standing (Sta) | Sit vs. H | Sta vs. H | Sta vs. Sit |
|---|---|---|---|---|---|---|
| Median [IQR] | 68.00 [66.00–78.00] | 71.00 [67.00–78.00] | 80.00 [80.00–82.50] | | | |

H: supine; Sit: sitting; Sta: standing; AOP: arterial occlusion pressure; BZ: biological zero; RHmax: reactive hiperemia; RT: recovery time; AVNN: Average NN Interval; SDNN: Standard Deviation of NN Intervals; LF/HF: Low Frequency/ High Frequency Ratio; HR: heart rate.

than supine (mean difference: 0.78 PU; p=0.003; r=−0.63). Standing resulted in significantly higher RHmax than both supine (mean difference: 12.7 PU; p=0.003; r=−0.63) and sitting (mean difference: 7.2 PU; p=0.003; r=−0.63). Sitting also showed significantly greater RHmax than supine (mean difference: 5.5 PU; p=0.006; r=−0.56).

Standing showed significantly greater time to peak than supine (mean difference: 11.0 s; p=0.003; r=−0.63) and sitting (mean difference: 6.2 s; p=0.003; r=−0.61). Sitting also presented significantly higher time to peak than supine (mean difference: 4.8 s; p=0.009; r=−0.56).

Fig 5 exhibits mean values and 95% confidence intervals for recovery time, Average NN Interval (AVNN); Standard Deviation of NN Intervals (SDNN), Low Frequency/ High Frequency Ratio (LF/HF); and heart rate (HR) across the three conditions.

Standing position demonstrated significantly greater TR compared to supine (mean difference: 10.11 min; p=0.003; r=−0.63) and sitting (mean difference: 8.11 min; p=0.003; r=−0.61). Sitting also had a significantly longer TR than supine (mean difference: 2.00 min; p=0.003; r=−0.63). Average NN intervals (AVNN) decreased significantly with upright posture. Both sitting (mean difference: −157.0 ms; p=0.003; r=−0.63) and standing (mean difference: −178.1 ms; p=0.003; r=−0.63) showed significantly lower AVNN values compared to supine. However, the difference between standing and sitting was not statistically significant (mean difference: −21.1 ms; p=0.159; r=−0.35).

Standing position demonstrated significantly higher SDNN than both supine (mean difference: 24.8 ms; p=0.003; r=−0.63) and sitting (mean difference: 5.8 ms; p=0.012; r=−0.53). Sitting also showed significantly greater SDNN compared to supine (mean difference: 19.0 ms; p=0.003; r=−0.63). The LF/HF ratio increased significantly with more upright positions. Standing position yielded significantly higher LF/HF values compared to both supine (mean difference: 0.68 A.U.; p=0.003; r=−0.62) and sitting (mean difference: 0.22 A.U.; p=0.003; r=−0.62). Sitting also showed significantly greater LF/HF than supine (mean difference: 0.46 A.U.; p=0.003; r=−0.63).

HR increased with upright posture. Standing resulted in significantly higher HR than both supine (mean difference: 11.0 bpm; p=0.006; r=−0.56) and sitting (mean difference: 10.0 bpm; p=0.006; r=−0.57). However, the difference between sitting and supine was not statistically significant (mean difference: 1.0 bpm; p=0.459; r=−0.26).

Exploratory analyses were conducted to examine whether the order in which sessions were performed (first, second, or third) influenced outcome measures. Kruskal–Wallis tests revealed no significant sequence effects for any variable: biological zero (BZ: H(2) = 2.76, p=0.251), time to peak (TP: H(2) = 1.41, p=0.494), recovery time (TR: H(2) = 2.89, p=0.236), AOPmin (H(2) = 0.54, p=0.765), AOP100 (H(2) = 2.13, p=0.345), AVNN (H(2) = 2.81, p=0.245), SDNN (H(2) = 2.85, p=0.240), LF/HF (H(2) = 2.61, p=0.271), and HR (H(2) = 0.67, p=0.716).

## 4. Discussion

This study demonstrated that body position significantly influences both microcirculatory hemodynamics and autonomic nervous system activity, providing new insights into the complex interplay between posture, post-occlusive reactive hyperemia (PORH), and autonomic regulation in healthy individuals. In line with our objective to investigate how varying lower limb positions affect vascular and HRV responses, we observed that standing posture elicited consistently greater responses across multiple vascular metrics—including increased resting flow, biological zero (BZ), reactive hyperemia

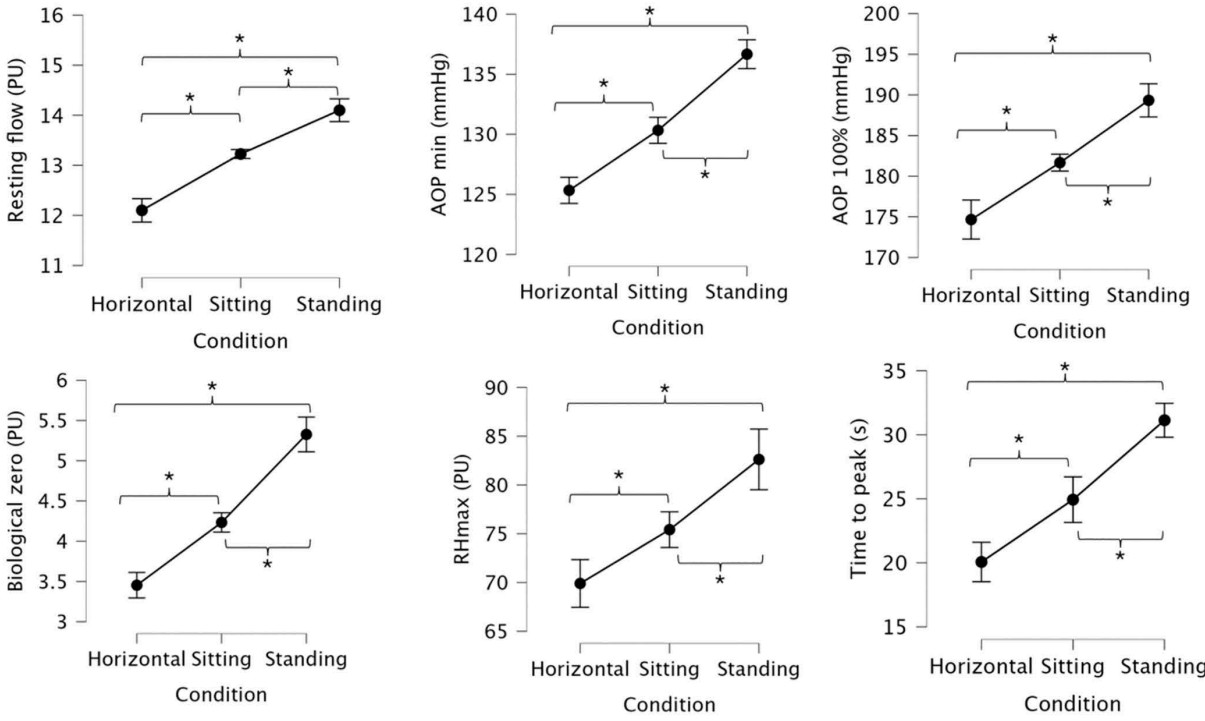

**Fig 4. Mean values and 95% confidence intervals for resting flow, arterial occlusion pressure (AOP), biological zero, reactive hyperemia (RHmax), and time to peak across the three conditions.** * indicates significant differences (p<0.05) between conditions.

(RHmax), and arterial occlusion pressures (AOP min and AOP 100%)—compared to sitting and supine positions. Additionally, recovery time (TR) and time to peak flow were significantly longer in more upright positions, suggesting altered reactive dynamics with postural elevation. Autonomic measurements mirrored these vascular shifts, with upright postures associated with reduced AVNN, increased heart rate, elevated SDNN, and a higher LF/HF ratio, indicating heightened sympathetic activity.

AOP, particularly AOPmin and AOP100%, as well as resting flow (RF), were significantly higher in seated and standing compared to supine positions. These posture-dependent changes are consistent with prior research showing that upright postures require greater cuff pressures to achieve arterial occlusion [44,45]. This effect can be explained by increased hydrostatic pressure in the dependent limb, which elevates transmural pressure and peripheral vascular resistance [46]. When upright, venous pooling reduces venous return and arterial compliance, requiring higher external cuff pressure to compress the vessel lumen. In addition, heightened sympathetic vasoconstrictor tone in upright positions further augments vascular resistance, compounding the increase in AOP [47]. Thus, the interaction of hydrostatic forces, reduced venous return, and autonomic adjustments provides a mechanistic explanation for the higher AOP values observed in upright postures [48,49].

In this study, biological zero (BZ), reactive hyperemia (RH), recovery time (TR), and time to peak (TP) demonstrated significant differences across the body positions tested (supine, seated, and standing). Biological zero (BZ), which reflects baseline perfusion during total occlusion [50], was significantly higher in the standing position compared to seated and supine conditions. This suggests that upright postures may influence the resting microcirculatory flow, potentially due to increased baseline vascular tone and gravity-induced shifts in blood distribution [51]. Similarly, the reactive hyperemia (RH) response was more pronounced in the standing position, which aligns with the literature indicating that standing

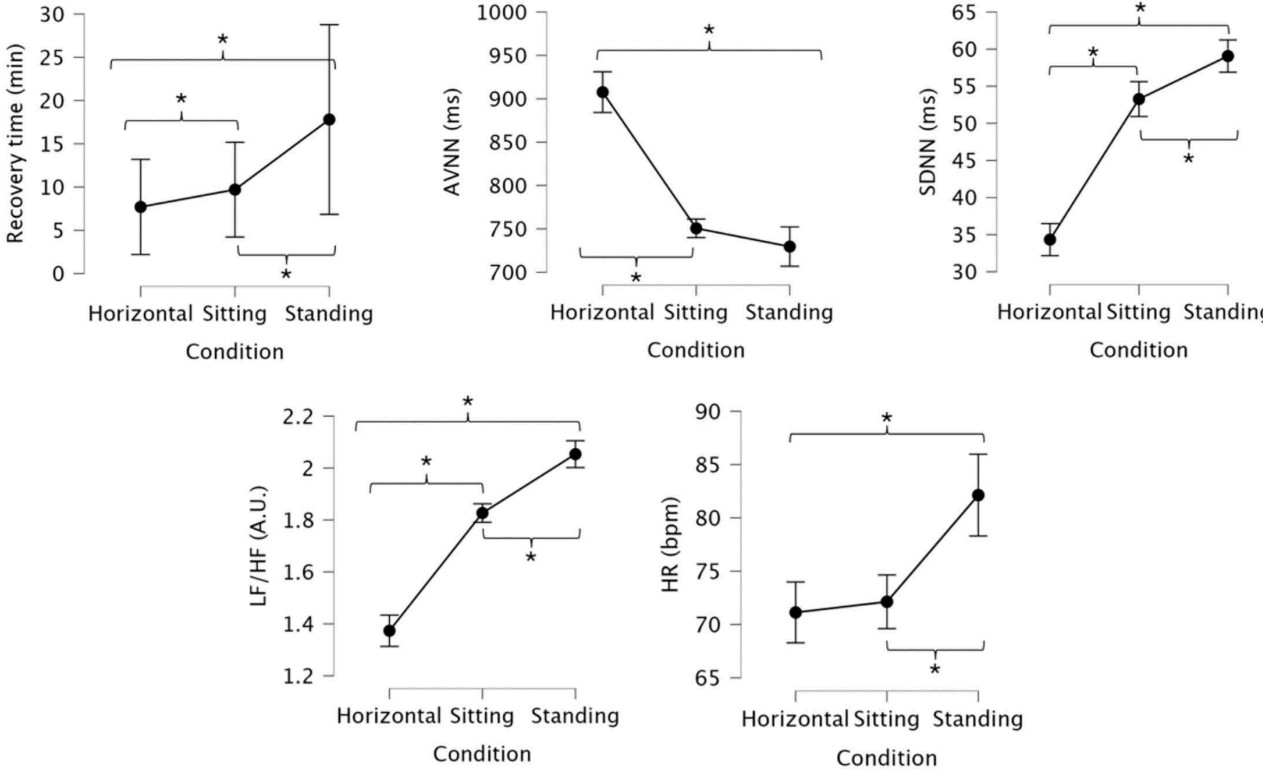

**Fig 5. Mean values and 95% confidence intervals for recovery time, Average NN Interval (AVNN); Standard Deviation of NN Intervals (SDNN), Low Frequency/ High Frequency Ratio (LF/HF); and heart rate (HR) across the three conditions.** * indicates significant differences (p<0.05) between conditions.

posture may lead to an augmented hyperemic response after arterial occlusion [52], likely due to increased venous return and sympathetic nervous system activation [53]. The greater RH in upright postures can be attributed to a combination of hydrostatic pressure and baroreflex-mediated autonomic adjustments [4]. Elevated hydrostatic pressure in the dependent limb augments the pressure gradient driving reperfusion, while sympathetic activation during orthostasis transiently constrains vasodilation, delaying but also amplifying the hyperemic rebound once occlusion is released [22]. These same mechanisms explain the longer time to peak (TP) and prolonged recovery time (TR) in standing, as increased vascular resistance and delayed clearance of metabolites extend the hyperemic phase.

In contrast, recovery time (TR) was substantially longer in the standing position, with the recovery process being slower compared to both supine and seated positions. This could be explained by the added vascular resistance and postural changes that the body experiences when upright, potentially delaying full restoration of blood flow to baseline levels [54]. The time to peak (TP), reflecting the time from occlusion release to the peak perfusion, was significantly longer in the standing condition, indicating a delayed hyperemic rebound. This delay in reaching peak perfusion may be due to the body's greater need to overcome gravitational forces in standing posture [51], leading to a more gradual perfusion recovery.

These findings are consistent with previous studies suggesting that body position significantly impacts vascular and microcirculatory responses during post-occlusive reactive hyperemia (PORH). Studies have shown that limb position relative to the heart affects the magnitude of reactive hyperemia, with greater blood flow responses observed when the limb is below the heart compared to above it [4,55]. This positional difference is partially mediated by mechanisms intrinsic to the

arterial wall [55]. In upright positions, increased hydrostatic pressure in the dependent limb promotes venous pooling and elevates vascular resistance, thereby delaying reperfusion and prolonging recovery time [56]. The release of occlusion induces a surge in shear stress, which activates endothelial vasodilators such as nitric oxide (NO) and endothelium-derived hyperpolarizing factor (EDHF), although this response may be attenuated by sympathetic vasoconstrictor activity [57,58]. In addition, exaggerated central hemodynamic responses in upright positions, including increased stroke volume and sustained heart rate elevation, contribute to greater cardiac output and perfusion pressure, resulting in a more pronounced and sustained hyperemic response compared to the supine position [17].

The results of heart rate variability (HRV) parameters—AVNN, SDNN, LF/HF, and HR—revealed significant changes across the three body positions (supine, seated, and standing). AVNN was significantly longer in the supine position compared to both seated and standing positions, suggesting greater parasympathetic dominance in the supine condition. This is consistent with previous studies indicating that lying down promotes increased parasympathetic activity and a lower heart rate, which is reflected in longer RR intervals [59]. Mechanistically, the supine position enhances venous return and baroreceptor loading, which sustain vagal outflow and limit sympathetic drive, thereby prolonging AVNN and stabilizing heart rate [32]. In contrast, HR was significantly higher in the standing position, corroborating findings that suggest an increase in sympathetic nervous system activity upon standing [60]. This response reflects baroreceptor unloading during orthostasis, which triggers vagal withdrawal and sympathetic activation to maintain arterial pressure and cerebral perfusion against gravitational stress [61]. The elevation in heart rate likely represents a compensatory mechanism to maintain cardiac output and blood pressure in response to the gravitational challenge of standing [62].

SDNN was significantly higher in both seated and standing positions compared to supine. This increase may indicate greater sympathetic modulation in these positions, likely due to the body's adaptation to the upright posture and the increase in autonomic nervous system activity required to maintain postural stability and blood pressure [63]. Interestingly, while SDNN was significantly greater in the seated and standing positions, it was still within the range of normal HRV values, suggesting that the body can maintain a healthy autonomic balance under these conditions.

The LF/HF ratio was significantly higher in the standing position compared to supine and seated. This finding points to an increase in sympathetic dominance when standing. In the current study, the significant increase in LF/HF ratio during the standing position supports the hypothesis that the body's autonomic regulation shifts towards sympathetic dominance in an upright posture, likely as a response to the need for increased circulation and blood pressure maintenance during standing [64]. Supine position promotes parasympathetic dominance, while standing activates sympathetic mechanisms to counteract gravitational challenges and maintain homeostasis [65].

Thus, the present findings demonstrate that posture exerts concurrent effects on both microcirculatory and autonomic function. Upright positions (sitting and standing) were characterized by higher resting flow, greater AOP, and prolonged recovery time, alongside a shift toward sympathetic predominance evidenced by shorter AVNN, higher LF/HF ratio, and increased HR. Conversely, the supine position favored lower AOP and stronger parasympathetic activity, consistent with shorter perfusion recovery and reduced vascular resistance. These parallel adaptations suggest that changes in hydrostatic pressure and venous return not only alter local PORH responses but also engage autonomic regulatory mechanisms. Such coupling between vascular and autonomic responses suggest the importance of considering both systems together when evaluating posture effects, particularly for optimizing blood flow restriction training protocols and ensuring cardiovascular safety.

An observation in our study was that upright postures elicited both enhanced reactive hyperemia (greater RHmax, delayed recovery time and time-to-peak) and autonomic shifts toward sympathetic predominance (shorter AVNN, increased LF/HF, elevated HR). These concurrent changes suggest a plausible physiological connection: sympathetic activation is known to attenuate hyperemic vascular responses, likely by opposing vasodilatory mechanisms. For example, in healthy males, acute sympathoexcitation induced by post-exercise circulatory occlusion markedly reduced vasodilatory response to passive leg movement—a surrogate for reactive [66]. Similarly, sympathetic stimulation has been shown

to blunt flow-mediated dilation of conduit arteries [67]. Furthermore, in patients with hypertension, greater sympathetic activation during reactive hyperemia has been associated with lower reactive hyperemia index (RHI) by tonometry [68]. Possibly sympathetic predominance in upright posture may contribute to delayed recovery following arterial occlusion in microvascular beds, aligning with the parallel changes observed in our HRV and RH measures.

Our study holds limitations that must be acknowledged. A limitation of the study is that sample size estimation was based on a medium effect size assumption rather than pilot or prior PORH-HRV data, due to the absence of directly comparable literature. Thus, replication with larger cohorts is needed to confirm the robustness of our findings. The acute, cross-sectional nature of the design also restricts conclusions about long-term physiological adaptations to postural changes. Additionally, while our methods taken PORH and HRV measures, the influence of potential confounding factors such as hydration status, circadian rhythm, or prior activity levels was not fully controlled. Another limitation is the choice of a diastolic blood pressure threshold of 95 mmHg for participant exclusion. Although clinical hypertension is typically defined as ≥140/90 mmHg, we adopted a slightly higher diastolic cut-off to account for measurement variability and to prioritize participant safety during arterial occlusion procedures. This conservative approach may have excluded some individuals with borderline but clinically normal diastolic values, potentially limiting generalizability.

Despite these limitations, our findings have important practical implications. The significant increases in AOP, reactive hyperemia, and sympathetic activity in upright postures suggest that standing and seated positions impose greater hemo-dynamic and autonomic stress during BFR exercise compared to the supine position. From a clinical perspective, this dual effect must be interpreted with caution. On one hand, the greater vascular and autonomic load in upright positions may enhance the metabolic and hemodynamic stimulus for training adaptations, which could be advantageous in athletic or rehabilitation contexts. On the other hand, such stress may elevate cardiovascular risk, particularly in individuals with hypertension, peripheral arterial disease, or other vascular impairments. Therefore, we recommend that AOP be mea-sured in the same posture in which BFR exercise is performed, and that lower relative pressures (e.g., 40–50% of AOP) be considered when training in upright positions to balance safety with efficacy. In clinical or frail populations, the supine position may provide a safer option by reducing vascular loading while still delivering the benefits of BFR training. Further-more, these findings highlight the importance of incorporating posture-specific considerations into vascular health assess-ments and the design of individualized BFR protocols for populations with compromised circulatory function.

From a safety perspective, these findings provide considerations for prescribing BFR exercise. Because AOP was significantly higher in upright positions compared to supine, using cuff pressures determined supine may underestimate actual AOP needed during upright exercise, potentially reducing training efficacy. Conversely, applying high cuff pressures based on supine measurements in upright settings may lead to excessive occlusion, discomfort, or cardiovascular strain. This risk highlights the importance of individualizing AOP in the specific training position. Indeed, recent guidelines recom-mend applying 40–80% of individual limb occlusion pressure (LOP) to optimize safety and effectiveness in BFR training protocols [69]. Our results further suggest that while the standing position may provide a stronger hemodynamic and metabolic stimulus—potentially enhancing training adaptations—this also imposes greater autonomic and vascular stress. In healthy or athletic populations, such stress may be tolerable and even desirable for maximizing adaptations. However, in older adults or individuals with chronic conditions such as hypertension, diabetes, or peripheral arterial disease, the additional sympathetic activation and elevated AOP in the standing position could increase cardiovascular risk. For these populations, seated or supine BFR exercise may represent a safer alternative, with more conservative cuff pressures recommended if upright training is pursued. Beyond exercise applications, posture-dependent differences in vascular and autonomic responses also bear implications for preoperative assessments and rehabilitation programming. For example, individuals with impaired vascular function—such as the elderly or those with peripheral arterial disease—may exhibit exaggerated hemodynamic and sympathetic reactions to upright posture, which could elevate cardiovascular risk during therapeutic interventions. Monitoring reactive hyperemia and HRV responses may help clinicians adjust BFR protocols or other interventions to individual vascular sensitivity and tolerance levels.

## 5. Conclusions

This study shows that body position plays a critical role in shaping both microvascular reactivity and autonomic nervous system activity. Standing and seated postures consistently elicited heightened vascular responses—including increased arterial occlusion pressure, reactive hyperemia, and delayed recovery dynamics—alongside greater sympathetic activation, compared to the supine position. These results stress the importance of postural context when evaluating vascular function and autonomic balance, particularly in applications such as BFR exercise where accurate assessment of arterial occlusion pressure is essential for safety and efficacy. By establishing that upright positions demand greater vascular and autonomic adjustments, our findings offer a physiological basis for adjusting BFR protocols and highlight the need to standardize body position in both research and clinical practice.

## Author contributions

**Conceptualization:** Robert Trybulski.

**Data curation:** Robert Trybulski.

**Investigation:** Robert Trybulski, Adrian Kużdzał, Wacław Kuczmik, Grzegorz Biolik, Magdalena Hagner Derengowska.

**Methodology:** Robert Trybulski, Adrian Kużdzał, Wacław Kuczmik, Grzegorz Biolik, Magdalena Hagner Derengowska.

**Writing – original draft:** Robert Trybulski, Adrian Kużdzał, Gabriel Stanica Lupu, Wacław Kuczmik, Grzegorz Biolik, Magdalena Hagner Derengowska, Jakub Taradaj.

**Writing – review & editing:** Robert Trybulski, Adrian Kużdzał, Gabriel Stanica Lupu, Wacław Kuczmik, Grzegorz Biolik, Magdalena Hagner Derengowska, Jakub Taradaj.

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
