## [Decision Letter · Decision Letter 0]

6 Aug 2025

Dear Dr. Trybulski,

Thank you for submitting your manuscript to PLOS ONE. After careful consideration, we feel that it has merit but does not fully meet PLOS ONE’s publication criteria as it currently stands. Therefore, we invite you to submit a revised version of the manuscript that addresses the points raised during the review process.

We look forward to receiving your revised manuscript.

Kind regards,

Yih-Kuen Jan, PhD

Academic Editor

PLOS ONE

Journal Requirements:

2. We note that you have selected “Clinical Trial” as your article type. PLOS ONE requires that all clinical trials are registered in an appropriate registry (the WHO list of approved registries is at https://www.who.int/clinical-trials-registry-platform/network/primary-registries and more information on trial registration is at http://www.icmje.org/about-icmje/faqs/clinical-trials-registration/). Please state the name of the registry and the registration number (e.g. ISRCTN or ClinicalTrials.gov) in the submission data and on the title page of your manuscript. a) Please provide the complete date range for participant recruitment and follow-up in the methods section of your manuscript. b) If you have not yet registered your trial in an appropriate registry, we now require you to do so and will need confirmation of the trial registry number before we can pass your paper to the next stage of review. Please include in the Methods section of your paper your reasons for not registering this study before enrolment of participants started. Please confirm that all related trials are registered by stating: “The authors confirm that all ongoing and related trials for this drug/intervention are registered”. Please see http://journals.plos.org/plosone/s/submission-guidelines#loc-clinical-trials for our policies on clinical trials.

3. In this instance it seems there may be acceptable restrictions in place that prevent the public sharing of your minimal data. However, in line with our goal of ensuring long-term data availability to all interested researchers, PLOS’ Data Policy states that authors cannot be the sole named individuals responsible for ensuring data access (http://journals.plos.org/plosone/s/data-availability#loc-acceptable-data-sharing-methods).

4. Please include captions for your Supporting Information files at the end of your manuscript, and update any in-text citations to match accordingly. Please see our Supporting Information guidelines for more information: http://journals.plos.org/plosone/s/supporting-information .

Reviewers' comments:

Reviewer's Responses to Questions

**Comments to the Author**

1. Is the manuscript technically sound, and do the data support the conclusions?

Reviewer #1: Yes

Reviewer #2: Partly

Reviewer #3: Yes

2. Has the statistical analysis been performed appropriately and rigorously?

Reviewer #1: Yes

Reviewer #2: No

Reviewer #3: Yes

3. Have the authors made all data underlying the findings in their manuscript fully available?

Reviewer #1: Yes

Reviewer #2: Yes

Reviewer #3: Yes

4. Is the manuscript presented in an intelligible fashion and written in standard English?

Reviewer #1: Yes

Reviewer #2: No

Reviewer #3: Yes

Reviewer #1: The authors recruited 20 healthy adults to evaluate how body position influences microcirculatory and autonomic responses. The results showed that body position influences both microcirculatory dynamics and autonomic nervous system responses.

1. Randomization was implemented for the order of position for each participant. Considering with limited sample size in this study, it would be helpful to also report the actual order at the end.

2. Inconsistency in the sample size. The power analysis reported 12, the statistical methods section reported 16, but the results section reported 15. Clarification on these inconsistencies is needed.

3. A nonparametric test was conducted due to the potential non-normality. However, the mean and standard deviation were reported in Table 1. It’s more common and appropriate to report median and IQR instead, while using a nonparametric test.

4. TR has an extremely large SD as shown in Table 1. This should be further examined, and a sensitivity analysis is warranted. Also, along this concern, it may be informative to add each data observation into the figures to show the variability of observations.

Reviewer #2: The quality of this manuscript needs to be improved.

1. What are the data for quantifying HRV?

2. The authors stated that “Studies suggest that HRV typically increases with parasympathetic activity in the supine position, contrasting with a more pronounced sympathetic response when upright”. What does “HRV typically increases” mean?

3. The description of method of determining the arterial occlusion pressure is unclear.

4. Regarding the statistical analyses:

1) the Wilcoxon signed-rank test is not the standard post hoc method for the Friedman test, potentially increasing false-positive risks.

2) With small samples (n=16, 15 or 12?), the normal approximation of Z-values may be inaccurate, casting doubt on the reliability of effect size r

5. The introduction and discussion sections lack clarity in organization and demonstrate insufficient depth. Addittionally, the reationship between changes in reactive hyperemia and changes in HRV was not discussed.

For the effects of postures on reactive hyperemia, the authors may refer the following papers.

Effect of Durations of Wheelchair Tilt-in-Space and Recline on Skin Perfusion Over the Ischial Tuberosity in People With Spinal Cord Injury, 2013, Archives of Physical Medicine and Rehabilitation

Skin blood flow dynamics and its role in pressure ulcers, 2013, Journal of Tissue Viability.

Reviewer #3: The authors examined the effects of three body positions (supine, seated, and standing) on post-occlusive reactive hyperemia (PORH) and autonomic nervous system activity in healthy individuals. The results demonstrated that the standing position increased arterial occlusion pressure (AOP) and microcirculatory responses, as well as altered heart rate variability (HRV). However, there are still some issues that might need to be clarified.

SPECIFIC COMMENT

Abstract

1. Line 34-35: Please define all abbreviations (AVNN, SDNN, LF/HF, and HR) at first mention.

2. Line 36: “…sensorsStanding…” should be “…sensors. Standing…”?

3. Line 38: What do “upright postures” mean? Do upright postures include standing and sitting positions? I recommend making terminology consistent.

Introduction

1. Again, I recommend making terminology consistent or clearly defining upright positions.

2. Line 62: Please define EDHF at first mention.

3. Line 74-79: I recommend elaborating more on HRV and autonomic response to justify the purpose of investigating HRV in the PORH protocol. The relationship between HRV and PORH should be described as well.

4. Line 82: “raising concerns about understanding…” Please clarify what concerns and why the results of this study are critical for optimizing Blood Flow Restriction protocols.

5. Line 91-97: It might not be necessary to describe the detailed outcome measures here. The authors might consider replacing this part with HRV measures and describing detailed outcome measures in the method section.

Methods

1. Line 134: Please provide a reference for McKay’s participant classification or clearly describe the criteria of level 1 classification.

2. Line 135: The normal range of ABI should be 0.9-1.4.

3. Line 140: Why did the study choose 95 mmHg as the diastolic blood pressure threshold? The criteria of hypertension are 140/90.

4. Line 171: Please provide the model of ultrasound and the type of transducer.

5. Line 174: What is the model for Doppler ultrasound?

6. Line 174: What is the model for the laser Doppler flowmeter?

7. Line 203: Again, please define the upright position or simply describe sitting or standing position.

8. Line 208: Please describe which artery was measured.

9. Line 212: How did the study determine the dominant leg?

10. Line 213-214: What were the two arteries?

11. Line 240-243: How did this study determine the value and the time of peak perfusion? How to identify that the resting flow is reduced? The raw LDF signals are noisy. Did you apply any filter or any processing?

12. Line 270: Why did this study decide to use the medium effect size to estimate sample size? I suggest using the data from a previous study to perform estimation.

13. Line 275: Although the sequence of conditions is random, the study still needs to evaluate and report the effect of the sequence on outcome measures.

14. Line 285: What test was used for assessing the normality of data?

Results

1. Line 299: A participant was excluded from the study. Please explain the reason for the dropout.

2. Line 321: What is horizontal position? Is it the supine position?

3. The information in Table 1 is also shown in Figures 4 and 5. The authors might consider removing the figures.

Discussion

1. The authors discuss each outcome in each paragraph. It might be better to discuss the effects of positions on PORH and HRV together to better understand the whole physiological changes in different positions more comprehensively.

2. The current discussion lacks physiological explanations for the findings. The authors might need to include some physiological mechanisms of PORH and HRV to interpret the results.

3. The safety of BFR and clinical implications should be discussed.

4. Line 391-404: The authors may minimize repeating the study findings and elaborate more on the physiological mechanisms of how vascular resistance affects AOP.

5. Line 405-415: Again, please elaborate on the physiological mechanisms.

6. Line 466-469: “The significant increases in…the supine position.” What clinical recommendation for BFR training do the authors give according to the current results? Is it a good thing to impose greater hemodynamics and autonomic stress during BFR?

7. How do you evaluate the effects of the standing position on BFR training? Any potential harm to individuals with chronic conditions or older adults?

**Do you want your identity to be public for this peer review?** For information about this choice, including consent withdrawal, please see our Privacy Policy

Reviewer #1: No

Reviewer #2: **Yes: ** Fuyuan Liao

Reviewer #3: **Yes: ** FU-LIEN WU

---

## [Author Response · Author response to Decision Letter 1]

9 Sep 2025

Reviewer #1:

The authors recruited 20 healthy adults to evaluate how body position influences microcirculatory and autonomic responses. The results showed that body position influences both microcirculatory dynamics and autonomic nervous system responses.

AUTHORS: Dear reviewer, thank you so much for your comment.

1. Randomization was implemented for the order of position for each participant. Considering with limited sample size in this study, it would be helpful to also report the actual order at the end.

AUTHORS: Dear Reviewer, thank you very much. We have added the following information to Section 2.5: “To control for potential order effects and ensure balanced exposure, the sequence of positions was randomly assigned using sealed opaque envelopes. Specifically, 5 participants completed supine first, followed by seated and then standing; 4 participants began with seated, then standing, and finally supine; and 6 participants started with standing, followed by supine, and then seated.”

2. Inconsistency in the sample size. The power analysis reported 12, the statistical methods section reported 16, but the results section reported 15. Clarification on these inconsistencies is needed.

AUTHORS: Dear reviewer, thank you so much. We have corrected the typos since we collected data from 15 participants (9 men and 6 women). The information has been updated in both the abstract and the statistical methods section. Regarding the power analysis, it suggested a sample size of 12, but we collected data from more participants to account for potential dropouts, which ultimately did not occur.

3. A nonparametric test was conducted due to the potential non-normality. However, the mean and standard deviation were reported in Table 1. It’s more common and appropriate to report median and IQR instead, while using a nonparametric test.

AUTHORS: Dear Reviewer, thank you very much. We have updated the table 1 and added dedicated rows for the median and IQR to better clarify the information for readers, as you recommended.

4. TR has an extremely large SD as shown in Table 1. This should be further examined, and a sensitivity analysis is warranted. Also, along this concern, it may be informative to add each data observation into the figures to show the variability of observations.

AUTHORS: We thank the reviewer for pointing out the high variability in TR. Sensitivity analysis using Tukey’s method identified one extreme outlier (104 min) in the standing condition. Excluding this outlier, the TR median was 11.90 [10.40–12.68] compared to 12.10 [10.40–12.85] with the full dataset. The overall conclusions remain unchanged: standing shows longer TR than sitting and horizontal, even when the extreme outlier is removed.

Reviewer #2: The quality of this manuscript needs to be improved.

AUTHORS: Dear Reviewer, thank you very much. We have made our best efforts to improve the quality of the manuscript, carefully addressing all comments and suggestions provided. We remain open to further improvements to ensure the highest possible quality.

1. What are the data for quantifying HRV?

AUTHORS: Dear reviewer, thank you so much. We have expanded point 2 of the section 2.6., as follows “HRV was recorded continuously using a Polar H10 chest strap (Polar Electro Oy, Finland) and stored via the HRV Logger application. R-R intervals were sampled at 1000 Hz and exported in raw format. Each participant underwent a standardized 5-minute recording in the designated body position (supine, seated, standing) following a 10-minute stabilization period. HRV serves as a non-invasive indicator of autonomic nervous system function, reflecting the balance between the sympathetic and parasympathetic nervous systems' influences on heart rate [21]. Body position can notably shift this balance, affecting HRV measurements. Studies suggest that HRV typically increases with parasympathetic activity in the supine position, contrasting with a more pronounced sympathetic response when upright. This has relevance in blood flow restriction training (BFRT) and its effects on hyperemic rebound [22]. Frequency-domain indices were calculated using Fast Fourier Transform (FFT) spectral analysis with Welch’s method (window length: 256 s, 50% overlap, Hanning window). All HRV indices were computed using Kubios HRV Premium (Kubios Oy, Finland), which is widely used for clinical and sports applications. The primary outcomes analyzed across positions were median [IQR] values for AVNN (mean of all normal-to-normal R-R intervals, representing overall heart period), SDNN (standard deviation of all NN intervals, reflecting total HRV), LF/HF (ratio of low-frequency (0.04–0.15 Hz) to high-frequency (0.15–0.40 Hz) power, indicating sympathovagal balance), and HR.”

2. The authors stated that “Studies suggest that HRV typically increases with parasympathetic activity in the supine position, contrasting with a more pronounced sympathetic response when upright”. What does “HRV typically increases” mean?

AUTHORS: Dear reviewer, thank you so much. We have improved as follows: “Studies suggest that specific HRV indices reflecting parasympathetic activity — such as higher AVNN, greater SDNN, and increased high-frequency (HF) power — are more pronounced in the supine position [22,23]. In contrast, upright posture elicits a shift toward sympathetic dominance, reflected in shorter AVNN, reduced HF power, increased LF/HF ratio, and higher heart rate [24].”

3. The description of method of determining the arterial occlusion pressure is unclear.

AUTHORS: We thank the reviewer for noting that our description of arterial occlusion pressure (AOP) measurement was unclear. We have revised the Methods section to provide a clearer step-by-step description “(...) Arterial Occlusion Pressure (AOP) was determined individually for each participant on the dominant leg. A 13-cm wide pneumatic cuff (Riester®, Germany) was positioned proximally at the inguinal crease. Using a 2D Doppler ultrasound probe placed over the posterior tibial and dorsalis pedis arteries, arterial flow was continuously monitored. Cuff inflation began at 0 mmHg and increased in 10 mmHg increments after an initial rise to 100 mmHg. At each step, pressure was held for 30 seconds to allow flow stabilization, while Doppler confirmed vessel patency [11]. The AOP was defined as the cuff pressure at which arterial flow was no longer detectable by Doppler. For analysis, we recorded both the minimal pressure reducing resting flow (AOPmin) and the pressure causing complete occlusion (AOP100). This individualized approach accounts for inter-subject differences in limb circumference, vessel depth, and posture, consistent with published BFR protocols. The study employed a standard 5-minute occlusion test on the dominant leg [29].”

4. Regarding the statistical analyses:

1) the Wilcoxon signed-rank test is not the standard post hoc method for the Friedman test, potentially increasing false-positive risks.

AUTHORS: We appreciate the reviewer’s concern and the opportunity to clarify our approach. Because our design involved within-subject repeated measures, the Wilcoxon signed-rank test is an established and widely accepted post hoc test following the Friedman procedure for pairwise comparisons. This choice is supported in the nonparametric statistics literature for related-samples designs (e.g., Gibbons & Chakraborti, 2010). To directly address the concern regarding potential false-positive risks, we took several additional precautions: (i) Bonferroni correction was applied to all Wilcoxon comparisons, and the adjusted p-values are those reported in the main tables; (ii) We further verified robustness by computing exact Wilcoxon p-values with Holm correction, which provides strong family-wise error control while avoiding reliance on large-sample approximations. These supplementary results confirmed that the overall pattern of significance was unchanged; (iii) Finally, we reported Kendall’s W as a global effect size for each Friedman test, which is robust in small-sample settings and demonstrated consistently large effects across outcomes.

2) With small samples (n=16, 15 or 12?), the normal approximation of Z-values may be inaccurate, casting doubt on the reliability of effect size r

AUTHORS: We thank the reviewer for this valuable observation. We agree that in small samples the normal approximation underlying the Wilcoxon Z statistic may reduce the accuracy of effect size r. To address this, we have revised the Statistical Methods section to clarify that r values are reported as descriptive indicators of magnitude only. In addition, we now report Kendall’s W as a robust global effect size for the Friedman test, which does not rely on the normal approximation and is well suited for small-sample nonparametric designs. Across our outcomes, Kendall’s W values ranged from 0.56 to 1.00, consistently indicating large effects, thereby reinforcing the robustness of our findings despite the modest sample size. These were the updates: “The results of the Friedman test indicated that there were significant differences between conditions in rest flow (χ²(2) = 30.000, p < 0.001, W = 1.00), AOPmin (χ²(2) = 28.526, p < 0.001, W = 0.95), AOP100% (χ²(2) = 29.000, p < 0.001, W = 0.97), biological zero (BZ) (χ²(2) = 30.000, p < 0.001, W = 1.00), reactive hyperemia (RHmax) (χ²(2) = 25.200, p < 0.001, W = 0.84), time to peak (χ²(2) = 25.864, p < 0.001, W = 0.86), recovery time (TR) (χ²(2) = 29.525, p < 0.001, W = 0.98), AVNN (χ²(2) = 23.333, p < 0.001, W = 0.78), SDNN (χ²(2) = 26.533, p < 0.001, W = 0.89), LF/HF ratio (χ²(2) = 29.525, p < 0.001, W = 0.98), and HR (χ²(2) = 16.877, p < 0.001, W = 0.56).”

5. The introduction and discussion sections lack clarity in organization and demonstrate insufficient depth. Addittionally, the reationship between changes in reactive hyperemia and changes in HRV was not discussed.

AUTHORS: Dear reviewer, thank you so much. The following was added: “An observation in our study was that upright postures elicited both enhanced reactive hyperemia (greater RHmax, delayed recovery time and time-to-peak) and autonomic shifts toward sympathetic predominance (shorter AVNN, increased LF/HF, elevated HR). These concurrent changes suggest a plausible physiological connection: sympathetic activation is known to attenuate hyperemic vascular responses, likely by opposing vasodilatory mechanisms. For example, in healthy males, acute sympathoexcitation induced by post-exercise circulatory occlusion markedly reduced vasodilatory response to passive leg movement—a surrogate for reactive [54]. Similarly, sympathetic stimulation has been shown to blunt flow-mediated dilation of conduit arteries [55]. Furthermore, in patients with hypertension, greater sympathetic activation during reactive hyperemia has been associated with lower reactive hyperemia index (RHI) by tonometry [56]. Possibly sympathetic predominance in upright posture may contribute to delayed recovery following arterial occlusion in microvascular beds, aligning with the parallel changes observed in our HRV and RH measures.”

6. For the effects of postures on reactive hyperemia, the authors may refer the following papers. Effect of Durations of Wheelchair Tilt-in-Space and Recline on Skin Perfusion Over the Ischial Tuberosity in People With Spinal Cord Injury, 2013, Archives of Physical Medicine and Rehabilitation Skin blood flow dynamics and its role in pressure ulcers, 2013, Journal of Tissue Viability.

AUTHORS: Dear reviewer, thank you so much. We have used both of them to expand our paragraph as follows: “From a clinical perspective, posture-dependent changes in PORH and vascular conduction are significant, particularly in rehabilitation and preoperative settings. For example, AOP tends to be lower in the supine position, raising concerns about underestimating occlusion thresholds during BFRT [9,16]. Research also suggests that limb position affects PORH magnitude, with horizontally or downward-oriented limbs showing stronger hyperemic responses, likely due to hydrostatic pressure effects [4,17]. Beyond limb positioning, posture and unloading strategies also influence cutaneous microcirculation. In individuals with spinal cord injury, durations of wheelchair tilt-in-space and recline significantly alter skin perfusion over pressure-bearing regions [18]. Likewise, reviews of skin blood flow dynamics emphasize how positional changes contribute to ischemia–reperfusion responses and tissue vulnerability [19].”

Reviewer #3: The authors examined the effects of three body positions (supine, seated, and standing) on post-occlusive reactive hyperemia (PORH) and autonomic nervous system activity in healthy individuals. The results demonstrated that the standing position increased arterial occlusion pressure (AOP) and microcirculatory responses, as well as altered heart rate variability (HRV). However, there are still some issues that might need to be clarified.

AUTHORS: Dear Reviewer, thank you very much for your thoughtful and pedagogical comments. They have greatly helped us improve and enhance the clarity of the current version.

SPECIFIC COMMENT

Abstract

1. Line 34-35: Please define all abbreviations (AVNN, SDNN, LF/HF, and HR) at first mention.

AUTHORS: Dear reviewer, thank you so much. We have updated: “average normal-to-normal interval (AVNN), standard deviation of normal-to-normal intervals (SDNN), low-frequency/high-frequency power ratio (LF/HF), and heart rate (HR)”

2. Line 36: “…sensorsStanding…” should be “…sensors. Standing…”?

AUTHORS: Dear reviewer, thank you so much. We have fixed: “(...) sensors. Standing (…)”

3. Line 38: What do “upright postures” mean? Do upright postures include standing and sitting positions? I recommend making terminology consistent.

AUTHORS: We thank the reviewer for pointing out this ambiguity. We have revised the abstract and the main text to clarify that upright postures refer specifically to sitting and standing positions, and we now use this terminology consistently throughout the manuscript.

Introduction

1. Again, I recommend making terminology consistent or clearly defining upright positions.

AUTHORS: Dear reviewer, thank you so much. We have updated in introduction as follows “These physiological shifts are especially relevant in BFRT protocols, which are commonly performed in supine, seated, or standing positions. In the present study, we refer to seated and standing collectively as ‘upright positions,’ to distinguish them from the supine posture.”

2. Line 62: Please define EDHF at first mention.

AUTHORS: Dear reviewer, thank you so much. Updated version: “PORH describes the surge in blood flow following the release of a temporary occlusion, serving as a marker for endothelial health. Its regulation involves multiple mechanisms, including the axonal reflex and endothelium-derived factors such as nitric oxide (NO) and endothelium-derived hyperpolarizing factor (EDHF)”

3. Line 74-79: I recommend elaborating more on HRV and autonomic response to justify the purpose of investigating HRV in the PORH protocol. The relationship between HRV and PORH should be described as well.

AUTHORS: Dear reviewer, thank you so much. The following was added: “HRV provides a non-invasive index of ANS balance, reflecting the dynamic interplay between sympathetic and parasympathetic inputs to the heart [20]. Given that the ANS also regulates vascular tone, particularly through sympathetic vasoconstrictor activity, HRV measures can provide indirect insight into microvascular responses such as PORH [21]. Previous work indicates that autonomic shifts influence endothelial function and cutaneous vasodilation, with sympathetic activation attenuating hyperemic responses and parasympathetic predominance facilitating more efficient recovery [22]. Therefore, evaluating HRV in parallel with PORH allows for a more comprehensive assessment of how posture influences both vascular and autonomic components of circulatory regulation.”

4. Line 82: “raising concerns about understanding…” Please clarify what concerns and why the results of this study are critical for optimizing Blood Flow Restriction protocols.

AUTHORS: Dear reviewer, thank you so much. This is the updated

---

## [Decision Letter · Decision Letter 1]

19 Oct 2025

Influence of Body Position on Microcirculatory and Autonomic Responses During Arterial Occlusion in Healthy Adults

PONE-D-25-32645R1

Dear Dr. Trybulski,

We’re pleased to inform you that your manuscript has been judged scientifically suitable for publication and will be formally accepted for publication once it meets all outstanding technical requirements.

Kind regards,

Yih-Kuen Jan, PhD

Academic Editor

PLOS ONE

Additional Editor Comments (optional):

Reviewers' comments:

Reviewer's Responses to Questions

**Comments to the Author**

Reviewer #1: All comments have been addressed

Reviewer #2: All comments have been addressed

Reviewer #3: All comments have been addressed

2. Is the manuscript technically sound, and do the data support the conclusions?

Reviewer #1: (No Response)

Reviewer #2: Yes

Reviewer #3: Yes

3. Has the statistical analysis been performed appropriately and rigorously?

Reviewer #1: (No Response)

Reviewer #2: Yes

Reviewer #3: Yes

4. Have the authors made all data underlying the findings in their manuscript fully available?

Reviewer #1: (No Response)

Reviewer #2: Yes

Reviewer #3: Yes

5. Is the manuscript presented in an intelligible fashion and written in standard English?

Reviewer #1: (No Response)

Reviewer #2: Yes

Reviewer #3: Yes

Reviewer #1: (No Response)

Reviewer #2: (No Response)

Reviewer #3: (No Response)

**Do you want your identity to be public for this peer review?** For information about this choice, including consent withdrawal, please see our Privacy Policy

Reviewer #1: No

Reviewer #2: No

Reviewer #3: **Yes: ** Fu-Lien Wu

---

## [Editor Report · Acceptance letter]

PONE-D-25-32645R1

PLOS ONE

Dear Dr. Trybulski,

I'm pleased to inform you that your manuscript has been deemed suitable for publication in PLOS ONE. Congratulations! Your manuscript is now being handed over to our production team.

Kind regards,

on behalf of

Dr. Yih-Kuen Jan

Academic Editor

PLOS ONE